# Impact of Three Different Algorithms for the Screening of SSc-PAH and Comparison with the Decisions of a Multidisciplinary Team

**DOI:** 10.3390/diagnostics11101738

**Published:** 2021-09-22

**Authors:** Valentin Coirier, Céline Chabanne, Stéphane Jouneau, Nicolas Belhomme, Alice Ballerie, Claire Cazalets, Vincent Sobanski, Éric Hachulla, Patrick Jégo, Alain Lescoat

**Affiliations:** 1Department of Internal Medicine and Clinical Immunology, Rennes University Hospital, F-35238 Rennes, France; nicolas.belhomme@chu-rennes.fr (N.B.); alice.ballerie@chu-rennes.fr (A.B.); claire.cazalets@chu-rennes.fr (C.C.); patrick.jego@chu-rennes.fr (P.J.); alain.lescoat@chu-rennes.fr (A.L.); 2Department of Cardiology, Rennes University Hospital, F-35238 Rennes, France; celine.chabanne@chu-rennes.fr; 3Department of Respiratory Diseases, Rennes University Hospital, F-35238 Rennes, France; stephane.jouneau@chu-rennes.fr; 4INSERM, EHESP, Institut de Recherche en Santé, Rennes 1 University, F-35000 Rennes, France; 5Service de Médecine Interne et Immunologie Clinique, Centre de Référence des Maladies Autoimmunes Systémiques Rares du Nord et Nord-Ouest de France (CeRAINO), CHU Lille. Inserm, U1286–INFINITE–Institute for Translational Research in Inflammation, Université de Lille, F-59000 Lille, France; vincent.sobanski@chru-lille.fr (V.S.); eric.hachulla@chru-lille.fr (É.H.); 6Institut Universitaire de France (IUF), F-75000 Paris, France

**Keywords:** systemic sclerosis, pulmonary hypertension, pulmonary arterial hypertension, screening, algorithm, multidisciplinary team meeting

## Abstract

Background: to compare three existing screening algorithms of pulmonary arterial hypertension (PAH) in systemic sclerosis (SSc) with the results of a multidisciplinary team (MDT) meeting from a tertiary center. Methods: we conducted a monocentric longitudinal study from 2015 to 2018. All patients with SSc according to LeRoy’s classification were eligible. Patients were excluded in the case of missing data required by any of the three screening algorithms. The algorithms were applied for each patient at inclusion. Right heart catheterization (RHC) was performed based on the MDT decision. MDT members were all blinded from the results of the three algorithms regarding RHC recommendations. The RHC recommendations of each algorithm were compared with the MDT decision, and the impact on diagnosis and management was evaluated. Results: 117 SSc patients were consecutively included in the study, and 99 had follow-up data over the three-year duration of the study (10 deaths). Among the 117 patients, the MDT suggested RHC for 16 patients (14%), DETECT algorithm for 28 (24%), ASIG for 48 (41%) and ESC/ERS 2015 for 20 (17%). Among the 16 patients who had RHC, SSc-PAH was diagnosed in seven. Among patients with an initial recommendation of RHC based on at least one algorithm but not according to the MDT meeting, no SSc-PAH was diagnosed during the three-year follow-up. Results were unchanged when the new 2018 definition of PAH was applied instead of the previous definition. Conclusion: a MDT approach appears interesting for the screening of SSc-PAH, with a significant reduction of RHC performed in comparison with dedicated algorithms. The specific relevance of a MDT for the management and follow-up of patients with RHC recommended by existing algorithms but with no PAH warrants further studies.

## 1. Introduction

Systemic sclerosis (SSc) is a rare chronic autoimmune disorder characterized by vasculopathy, inflammation and fibrosis of skin and internal organs [1,2,3,4]. Currently, pulmonary arterial hypertension (PAH) is a major cause of SSc-related death [5,6]. The prevalence of PAH ranges from 5% to 20% [7,8] in patients with SSc. However, the onset of PAH is insidious, leading to a significant delay between the first clinical symptom and the final diagnosis of PAH [9]. Although earlier treatment leads to better prognosis [10], SSc-PAH still has a major impact on survival [11]. Reducing the time to diagnosis of this impactful complication is, thus, important and justifies a systematic screening in SSc patients [12].

Over the past decade, several algorithms and guidelines have been proposed (Figure 1) to improve screening and diagnosis. Hachulla et al. first compared early PAH diagnosis using transthoracic echocardiography (TTE) prior to referral for right heart catheterization (RHC) with routine clinical practice [13]. This strategy allowed the identification of patients with milder forms of SSc-PAH, and significantly improved the 8-year survival rate (64% in the early group, compared with 17% in the routine group, *p* = 0.0037) [10]. In 2012, the Australian Scleroderma Interest Group (ASIG) proposed a screening algorithm for SSc-PAH based on the results of pulmonary function tests (PFT) and N-terminal pro-B-type natriuretic peptide (NT-pro-BNP) levels. In this algorithm, TTE was proposed only if the first step was positive, before performing RHC. The reported sensitivity was 94.1%. Coghlan et al. developed, in 2014, the first evidence-based algorithm for screening of SSc-PAH [14]. This approach, called the DETECT algorithm, was based on multiple parameters (clinical variables, PFT, immunological, biological, electrocardiographic and TTE), and offered a very high sensitivity (96%) for the screening of PAH-SSc. The European Society of Cardiology (ESC) and the European Respiratory Society (ERS) also regularly update guidelines for the diagnosis and the management of pulmonary hypertension (PH), with a last update in 2015 and a specific focus on connective tissue diseases (CTD)-associated PAH [15]. Moreover, a new definition of PAH was proposed in 2018 [16], defining PAH as the association of a mean pulmonary arterial pressure (mPAP) > 20 mmHg and a pulmonary vascular resistance (PVR) ≥ 3 Wood units (WU). A recent American single center study demonstrated that the sensitivity of DETECT was higher than the ESC/ERS guidelines but subsequently led to a higher number of RHCs and a high rate of false-positive, unnecessary RHC, although this could be considered an appropriate result for a screening strategy, in contrast with a diagnostic approach.

Multidisciplinary team (MDT) discussion could be helpful for the screening and diagnosis of rare and complex disorders such as SSc-PAH and may help to spare invasive assessments such as RHC. For example, according to the ATS/ERJ/JRS/ALAT 2018 guidelines, a MDT working group is recommended to discuss the need and relevance of invasive lung biopsy for the diagnosis of idiopathic pulmonary fibrosis (IPF), as some characteristic high-resolution computed tomography (HRCT) features may sometimes be missing [17]. This MDT discussion has thus become the cornerstone for the diagnosis of IPF, decreasing the number of lung biopsies performed. Similarly, such a MDT-based approach could be beneficial for the diagnosis of SSc-PAH. As SSc-PAH is among the leading causes of SSc-related death, the early diagnosis of SSc-PAH is of high importance. The relevance of MDT in comparison with other screening strategies has never been evaluated to date. To explore this question, we conducted a longitudinal study comparing the baseline number of RHCs recommended by the three existing algorithms with that recommended by a MDT in a tertiary center and evaluated the baseline prevalence and subsequent incidence of SSc-PAH during a 3-year follow-up to discuss the relevance of each approach.

## 2. Materials and Methods

### 2.1. Patients and Follow-Up

From November 2014 to May 2015, all consecutive adults admitted to an outpatient clinic for annual assessment of SSc-related visceral complications were included in this observational study. The exclusion criterion was missing data to apply any of the three selected PAH algorithms (DETECT, ASIG or ESC/ERS 2015 guidelines). The diagnosis of SSc was based on Le Roy and Medsger’s classification criteria for early SSc [18]. Patients already diagnosed with SSc-PAH were not excluded, to test the sensitivity of the different algorithms. No power analysis was performed to assess the sample size. All patients who met the inclusion criteria during the study period were included.

During the follow-up period, in accordance with routine care in our center, an annual SSc visit dedicated to visceral assessment was performed with physical examination, blood sampling, PFTs, chest X-rays or HRCT, electrocardiogram and systematic TTE. When a death occurred during the follow-up period, the etiology was defined based on the patient’s medical record and based on a phone call to her/his general practitioner.

### 2.2. PAH Algorithms

The three different algorithms (DETECT, ASIG and ESC/ERS 2015) were applied at baseline for each patient. The clinician in charge of the patient was blinded to the results of each algorithm. All algorithms were performed a second time, after 3-year follow-up, for comparison with baseline. To apply the ASIG algorithm [19], NT-pro-BNP, predicted diffusing capacity for carbon monoxide (DCLO) and forced vital capacity (FVC) were necessary. This algorithm is composed of two components: (1) Component A is positive if NT-pro-BNP is above 210 pg/mL; (2) Component B is positive if DLCO is lower than 70% of predicted and the FVC/DLCO ratio is above 1.8. When at least one of the two components was positive, RHC was recommended. The DETECT algorithm [14] was applied using the online calculator [20] according to the application criteria: patients >18 years with a diagnosis of SSc of >3 years’ duration from the first non-Raynaud’s phenomenon (RP) symptom and a DLCO <60% of predicted. Six non-TTE variables were required to calculate the step 1 risk score. If one of them was missing, the calculator could still be used. If the step 1 risk score was above 300, TTE was recommended. To calculate the risk score for step 2, TTE-based right atrium area and tricuspid regurgitation velocity (TRV) were required. If tricuspid regurgitation was not detectable, this could be specified and was not counted as a missing datum per se. If the step 2 risk score was above 35, RHC was recommended.

The ESC/ERS 2015 guidelines [15] recommend an annual screening for all SSc-patients, with TTE, DLCO and NT-pro-BNP measurement. Based on TTE findings, RHC was recommended for patients with intermediate or high risk for PAH. The multidisciplinary team and physician in charge of the patient were blinded from the results of all three algorithms at baseline and during follow-up.

### 2.3. Multidisciplinary Team (MDT)

In our tertiary center, in accordance with French guidelines [21], all patients with SSc benefited from a biannual physical examination and an annual PAH evaluation (including the results from an annual visceral assessment as described above [22]). After these SSc-dedicated evaluations, a monthly MDT meeting including a panel of expert clinicians in PH assessment (internists, rheumatologists, cardiologists, pulmonologists and radiologists) reviewed the patient’s results (blinded from the results of all algorithms at baseline and during follow-up) and recommended RHC or not.

### 2.4. PAH Definitions

As was recommended until 2018 [13], SSc-PAH diagnosis was confirmed for RHC when mPAP was≥ 25 mmHg, with pulmonary capillary wedge pressure (PCWP) ≤ 15 mmHg, in the absence of extensive interstitial lung disease (ILD) [15]. When PCWP was >15 mmHg, PH secondary to left heart disease was diagnosed. In February 2018, the Sixth World Symposium on Pulmonary Hypertension (WSPH) revised this definition of PAH [16], with a new suggested mPAP threshold of 20 mmHg associated with PVR ≥ 3 WU for the diagnosis of PAH. In this study, we prospectively applied the first definition of PAH and retrospectively applied the new 2018 definition.

### 2.5. Data Collection

All data required by the three algorithms (Figure 1) were collected at baseline and at the 3-year follow-up visit. In the interval, when RHC was recommended by MDT, data required by the three algorithms were also collected again. Informed consent of all patients was obtained, and a declaration at the Commission Nationale de l’Informatique et des Libertés was obtained (declaration 1980161 v0).

### 2.6. Statistics

Quantitative variables were expressed by the median (25th percentile–75th percentile), and comparisons between groups were performed using a Mann–Whitney U test or Kruskall–Wallis test. Categorical variables were compared with a chi-square test or Fisher’s exact test when appropriate. Kaplan–Meier curves were estimated for 3-year survival, with differences between SSc-PAH and no SSc-PAH groups assessed by the log rank test. A *p*-value lower than 0.05 was considered statistically significant.

## 3. Results

### 3.1. Population Characteristics at Baseline

From November 2014 to May 2015, 207 unselected, consecutive SSc patients had an annual SSc visit. Ninety of them were excluded due to missing data required for one of the three algorithms, and 117 patients were finally included (Figure 2). The median age was 55 (46–66) years old, with a majority of females (sex ratio = 0.3). One hundred and five patients (90%) fulfilled the 2013-ACR/EULAR criteria for SSc [23], and all of them fulfilled the 2001 classification criteria for early SSc. The median disease duration (from the first non-RP symptom) was 6 (3–11) years. Main clinical, biological and functional characteristics are detailed in Table 1.

### 3.2. Recommendations According to Algorithms and MDT Based on Baseline Data

DETECT was not applied to 79 of the patients, because they had DLCO ≥ 60% (*n* = 68) and/or a RP duration ≤ 3 years (*n* = 32). According to the DETECT algorithm at baseline, TTE was recommended for 92/117 patients (79%) and RHC for 28/117 patients (24%) (Figure 3). Among these 28 patients, 54% had ILD on HRCT, TRV was 2.8 m/s (2.6–3.2), FVC was 93% (79–107%) and DLCO 50% (32–55%) of the predicted value (Table 2). According to ESC/ERS 2015 guidelines, RHC was recommended for 20 of 117 patients (17%); among them, 45% had ILD, TRV was 3.1 m/s (2.9–3.4), FVC was 90% (77–101%) and DLCO 47% (30–64%). The ASIG algorithm recommended RHC for 48 of 117 patients (41%). Forty percent of them had ILD, TRV was 2.8 m/s (2.5–3), FVC was 98% (82–116%) and DLCO 53% (40–64%).

Among the 117 patients included, MDT proposed RHC for 16 patients (14%). In these patients, NT-pro-BNP was 323 pg/mL (101–1125), 56% had ILD, TRV was 3.2 m/s (2.9–3.4), FVC 90% (79–101%) and DLCO 39% (30–57%). When RHC was suggested by the MDT or according to ESC/ERS 2015 guidelines, TRV was significantly higher compared to the DETECT and ASIG groups (*p* = 0.002). No other characteristics differed between groups defined by the different algorithms (Table 2).

### 3.3. SSc-PAH Diagnosis

Sixteen RHC operations were finally performed, according to the MDT decisions. SSc-PAH was diagnosed in seven patients (Figure 3), post-capillary PH in two patients and seven RHC patients showed no PH as defined prior to 2018 (Table 3). In comparison with patients without PAH, those with SSc-PAH had more frequently positive anti-centromere antibodies (86% versus 22%, *p* = 0.041) and higher FVC/DLCO ratios (2.63 versus 2.07, *p* = 0.012). DLCO was numerically lower in the PAH group (31% versus 54%, *p* = 0.071), as was pulse oximetry at the end of the six-minute walk test (88% versus 96%, *p* = 0.067). All three algorithms would have recommended RHC for patients in whom SSc-PAH or post-capillary PH had been diagnosed; thus, no patient would have been missed by the algorithms. For 3/9 (33%) and 2/9 (22%) patients with normal RHC, the indication was not recommended by the ESC/ERS 2015 guidelines and ASIG/DETECT algorithms, respectively.

Each symbol represents one patient (*n* = 117). Those for whom right heart catheterization (RHC) was recommended by DETECT (*n* = 28) are in the blue circle, by ESC/ERS 2015 (*n* = 20) in the red circle and by ASIG (*n* = 48) in the yellow one. Patients for whom the MDT indicated RHC are represented by squares (*n* = 16). Symbols are grey for patients without indication of RHC (*n* = 60), black when at least one algorithm retained RHC indication (*n* = 57) and red when pulmonary arterial hypertension was confirmed by RHC (*n* = 7).

### 3.4. Three-Year Follow-Up

In 2018, 99 of the 117 patients (85%) were still being followed up in our department. Among these 117 patients, ten died during the study period, three of whom had SSc-PAH. The 3-year survival rate was significantly different between patients with SSc-PAH and those without (Appendix A).

Among these 99 patients, a new recommendation of RHC during the study period was suggested by the MDT for four of them. Among those, DETECT and ASIG algorithms were already positive in 2015 for one patient, as were ASIG and ESC/ERS 2015 guidelines for another one. For the two other patients, all algorithms were negative in 2015. RHC was performed for these four patients, and post-capillary PH was diagnosed for one patient (ASIG and ESC/ERS positive at inclusion, not DETECT), but there was no PAH.

In 2018, 22 patients for whom the DETECT algorithm recommended RHC at baseline were still being followed up in our center. Three years later, RHC was still recommended only for 15 of them (68%). This rate was, respectively, 7/14 (50%) and 29/35 (83%) for the ESC/ERS 2015 and ASIG algorithms.

### 3.5. Impact of the New PH Definition

According to the new suggested mPAP threshold of 20 mmHg, there was no additional diagnosis of PAH-SSc for patients in whom RHC was performed. Two patients had a mPAP between 20 mmHg and 25 mmHg, but PVR did not reach 3 WU.

## 4. Discussion

The need to improve the early diagnosis of SSc-PAH led to the development of screening algorithms during the last decade. This study compares the three main algorithms available to date—ASIG, DETECT and ESC/ERS 2015 guidelines—and compares their recommendations with the suggestions of a MDT of SSc experts from a tertiary center, blinded from the results of each algorithm. This MDT approach provides an original and practical method for the screening of SSc-PAH. The MDT meeting ensured a comparison between the viewpoint of the clinician in charge of the patient, the review of TTE by a cardiologist specialized in PAH, the interpretation of HRCT by a radiologist and a pulmonologist. Early diagnosis of SSc-PAH remains challenging despite the progress achieved over the past 20 years [13]. The use of algorithms is helpful to clinicians, but they may lack specificity, leading to a high number of RHCs that can be at risk of adverse events [24]. Limiting the number of RHCs may also spare money and time [25].

Our study confirms the high sensitivity of the DETECT algorithm, as already reported in previous studies [25,26,27,28]. There did not seem to be any PAH that went undiagnosed by DETECT in our work. One of the strengths of this algorithm is the inclusion of multiple variables (clinical, biological (anti-centromere antibodies) and echocardiographic items). As in the results from the DETECT study, in our study anti-centromere antibodies were more frequent in patients with SSc-PAH compared to those without. However, the number of RHCs recommended by DETECT was much higher than that recommended by the MDT or ESC/ERS guidelines. The number of TTEs recommended was not much lower than what it would be if TTE were systematically performed annually. The interval at which the algorithm should be repeated, and what to do when it is positive but with no PAH on RHC in the end, are still pending issues. A recent study suggested that in the case of RHC recommended by DETECT but with no PAH on RHC, follow-up should be based on the other available algorithms.

The ASIG algorithm had the highest number of recommended RHCs (41% of the patients), whereas it was previously described as having a better specificity than DETECT [26]. The applicability of this algorithm to a European population may, thus, be an issue and it may not be relevant, especially once DETECT and a first RHC have ruled out PAH. The use of this algorithm theoretically makes TTE or HRCT non-systematic, but these evaluations are, however, helpful for the screening of other SSc-related complications [21].

The ESC/ERS 2015 guidelines had a lower recommendation of RHC (17% of the patients). The previous comparisons of ESC/ERS guidelines 2009 with other algorithms reported a worse sensitivity for the screening of SSc-PAH. In 2015, the new guidelines lowered the threshold to recommend RHC in at-risk populations (including CTD), but DETECT retains a higher sensitivity despite questionable specificity regarding the number of recommended RHCs. In our study, using the MDT as reference standard, these updated guidelines recommended RHC for all patients who had PAH at the end of and/or during follow-up. For the recommendation of RHC, the MDT seems an interesting approach, allowing a thoughtful discussion between different experts. A MDT also allows the discussion of diagnostic alternatives without limiting the question to a dichotomic approach solely assessing the absence or presence of PH. Such an approach could, thus, be especially relevant where DETECT recommends RHC but with no diagnosis of PAH in the end, a scenario which suggests that alternative diagnoses should be ruled out and that a coordinated follow-up is needed to discuss repeated RHC, as DETECT cannot be used in this situation anymore. In our study, the MDT approach resulted in the lowest number of RHCs, although the assessment of sensitivity and specificity of the MDT was impossible, as RHC was not systematically performed in all patients, in contrast to the initial DETECT study. This was a major limitation of our study, but the absence of new clinical/biological/echocardiographic worsening during the 3-year follow-up period suggests that no PAH was missed.

Acknowledging that a mPAP between 20 mmHg and 25 mmHg is associated with an increased risk of morbidity, mortality [29,30,31] or PAH development [32], and that normal resting mPAP is within the range of 14–17 mmHg [33], a new definition of PAH was proposed in 2018 [16]. This new definition now includes PVR ≥ 3 WU as a diagnostic condition, whereas it had been unnecessary in the previous PAH definition. Using the updated PAH definition, Young et al. [27] recently confirmed a sensitivity of 1 for the DETECT algorithm and of 0.8 for the ESC/ERS 2015 guidelines, even for patients with DLCO > 60% predicted. However, the reported specificity of DETECT decreased from 0.33 to 0.29 because of the application of the 2018 PAH definition. This low impact of the revised PAH definition was also supported by Jaafar et al. in 2019 from the same center [34]. In their monocentric retrospective study, the authors compared the PAH prevalence in two SSc-patient cohorts according to the previous or the updated definition. Their analysis revealed that, among 131 SSc patients without PAH according to the previous definition, four reached the pre-capillary PH criteria for the new definition and only one was finally diagnosed with PAH. Thus, the updated PH definition did not appear to have a major impact on the diagnosis of SSc-PAH. Available data seem to show that the association of PVR ≥ 3 WU with mildly elevated mPAP (between 21 and 25 mmHg) is a rare condition in SSc patients [35]. In our work, applying this new PAH definition had no impact, as, among patients with RHC, two patients had a mPAP between 20 mmHg and 25 mmHg, but PVR did not reach 3 WU. A recent study challenged the 3 WU threshold by showing an increased risk of mortality and heart failure in patients with PVR ≥ 2.2 WU [36]. The studied population was, however, mainly composed of patients with a history of heart failure and chronic obstructive pulmonary disease, which is not representative of SSc patients.

Our study had several limitations, mainly its single-center design and the absence of systematic RHC, which is the reference standard to assess the strength of a screening strategy. To our knowledge, systematic RHC has been performed in only two studies [14,27] that were dedicated to the DETECT algorithm. In our study, among patients with baseline recommendation of RHC according to DETECT or ASIG, none showed clinical/biological/echocardiographic worsening during the 3-year follow-up period, suggesting that no PAH was missed by the MDT, although the absence of systematic RHC precludes firm conclusions. A MDT meeting must gather PAH and scleroderma specialists, who inevitably differ in terms of experience and expertise from one center to another. This was another limitation of this approach. However, guidelines for the management of SSc patients suggest that follow-up, organ assessment and related therapeutic decisions should rely on centers with an expertise in SSc management [37]. This expert multidisciplinary approach is also currently used for other disorders, such as IPF [17], and appears to be an interesting approach to determine the need for lung biopsy and the final diagnosis of ILD.

In conclusion, a multidisciplinary approach for the screening of SSc-PAH leads to a significant reduction of RHC performed, as compared to existing algorithms, but systematic RHC is needed to fully explore the sensitivity of MDTs. Considering their high sensitivity, SSc-PAH screening algorithms such as DETECT are helpful in limiting the number of RHCs when they do not recommend RHC. The high sensitivity of DETECT results from the higher number of recommended RHCs as compared to the 2015 ERS/ERC guidelines. A MDT could allow discussion of alternative diagnoses with a practical discussion of the relevance of repeating RHC after a first RHC without PAH, as DETECT has neither been evaluated nor designed for this situation. The specific relevance of MDTs for the management and follow-up of patients with RHC recommended by DETECT but with no PAH warrants further studies.

## Figures and Tables

**Figure 1 diagnostics-11-01738-f001:**
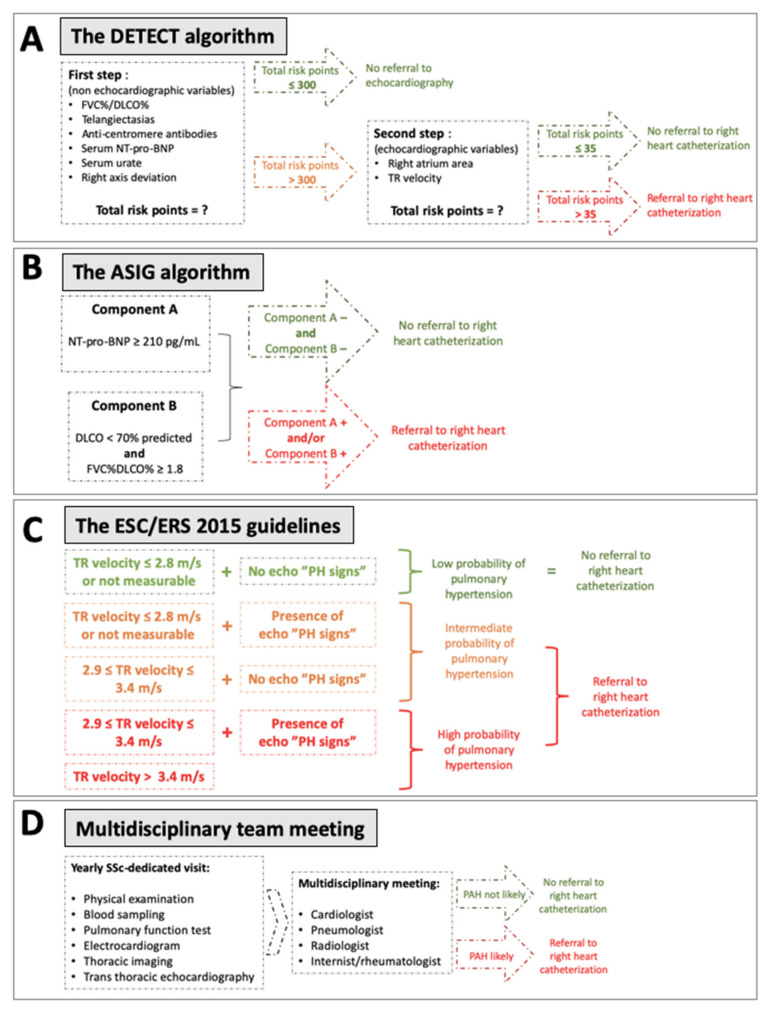
Description of the four different screening situations compared. (**A**) DETECT algorithm; (**B**) ASIG algorithm; (**C**) ESC/ERS 2015 guidelines; (**D**) Multidisciplinary team meeting. DLCO: diffusing capacity for carbon monoxide; FVC: forced vital capacity; NT-pro-BNP: N-terminal pro-B-type natriuretic peptide; PAH: pulmonary arterial hypertension; PH: pulmonary hypertension; TR: tricuspid regurgitation.

**Figure 2 diagnostics-11-01738-f002:**
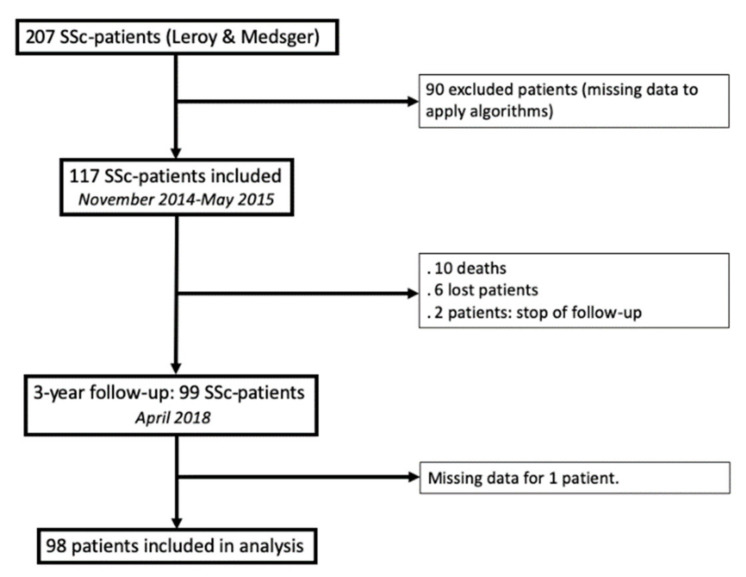
Flow chart.

**Figure 3 diagnostics-11-01738-f003:**
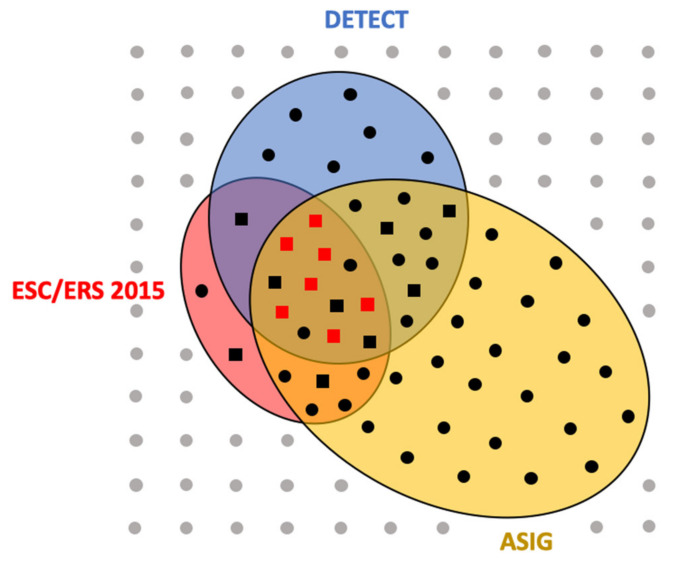
Representation of right heart catheterization indications according to the three algorithms and the multidisciplinary team discussion.

**Table 1 diagnostics-11-01738-t001:** Population characteristics at baseline and at the end of follow-up, 3 years later.

*n* (%) or Median (IQR)	Baseline (*n* = 117)	End of FU (*n* = 99)
Age at diagnosis (years old)	55 (46–66)	54 (46–63)
Disease duration (years)	6 (3–11)	9 (6–15)
Sex ratio	0.3	0.29
ACR/EULAR 2013 criteria	105 (90%)	89 (90%)
Telangiectasia	83 (71%)	67 (68%)
Right axis deviation on ECG	8 (7%)	5 (5%)
Positive anti-centromere antibody	65 (56%)	56 (57%)
NT-pro-BNP (pg/mL)	94 (50–164)	103 (59–173)
Serum urate (mmol/L)	274 (238–357)	270 (224–345)
DLCO (% of predicted value)	64 (52–72)	63 (53–72)
FVC/DLCO ratio	1.68 (1.44–1.86)	1.63 (1.41–1.84)
Interstitial lung disease	46 (39%)	40 (40%)
Estimated sPAP (mmHg)	25 (20–30)	30 (26–35)
Right atrium surface (cm^2^)	14 (12–16)	13 (12–15)
Death during the study period		10/117 (9%)

ACR: American College of Rheumatology; DLCO: diffusing capacity for carbon monoxide; ECG: electrocardiogram; FVC: forced vital capacity; FU: follow-up; NT-pro-BNP: N-terminal pro-B-type natriuretic peptide; sPAP: systolic pulmonary arterial pressure.

**Table 2 diagnostics-11-01738-t002:** Population characteristics according to the results of algorithms at baseline.

*n* (%) or Median (IQR)	DETECT + *n* = 28/117	ESC 2015 + *n* = 20/117	ASIG + *n* = 48/117	MDT Decision + *n* = 16/117	*p*-Value
Age at diagnosis	56 (47–65)	64 (53–70)	64 (54–68)	64 (53–70)	0.297
Disease duration	9 (6–14)	9 (4–19)	6 (4–13)	7 (4–19)	0.562
Sex ratio	0.22	0.43	0.37	0.23	0.786
ACR/EULAR 2013 criteria (*n* [%])	27 (96%)	19 (95%)	45 (94%)	15 (94%)	0.970
Telangiectasia (*n* [%])	27 (96%)	19 (95%)	40 (83%)	15 (94%)	0.213
Right axis deviation on ECG	4 (14%)	4 (20%)	5 (10%)	3 (19%)	0.757
Anti-centromere antibody (*n* [%])	15 (54%)	12 (60%)	26 (54%)	8 (50%)	0.944
NT-pro-BNP (pg/mL)	137 (97–390)	318 (126–705)	240 (77–497)	323 (101–1125)	0.361
Serum urate (mmol/L)	297 (241–416)	354 (297–476)	297 (238–416)	357 (297–520)	0.081
DLCO (% of predicted value)	50 (32–55)	47 (30–64)	53 (40–64)	39 (30–57)	0.178
FVC (% of predicted value)	93 (79–107)	90 (77–101)	98 (82–116)	90 (79–101)	0.160
FVC/DLCO ratio	2.12 (1.73–2.52)	2.16 (1.53–2.62)	2.02 (1.81–2.40)	2.28 (1.70–2.62)	0.744
Interstitial lung disease on HRCT (*n* [%])	15 (54%)	9 (45%)	19 (40%)	9 (56%)	0.553
Tricuspid regurgitation velocity(m/s)	2.8 (2.6–3.2)	3.1 (2.9–3.4)	2.8 (2.5–3)	3.2 (2.9–3.4)	0.002 *
Estimated sPAP (mmHg)	35 (25–48)	45 (38–50)	30 (25–44)	47 (36–50)	0.002 *
Right atrium surface (cm^2^)	15 (12–19)	16 (14–21)	14 (12–19)	17 (14–22)	0.194
DETECT +	28 (100%)	13 (65%)	21 (44%)	14 (88%)	NA
ASIG +	21 (75%)	17 (85%)	48 (100%)	14 (88%)	NA
ESC 2015 +	13 (46%)	20 (100%)	17 (35%)	13 (81%)	NA
MDT decision +	14 (50%)	13 (65%)	14 (29%)	16 (100%)	NA

ACR: American College of Rheumatology; DLCO: diffusing capacity for carbon monoxide; ECG: electrocardiogram; FVC: forced vital capacity; HRCT: high-resolution computed tomography; MDT: multidisciplinary team; NT-pro-BNP: N-terminal pro-B-type natriuretic peptide; sPAP: systolic pulmonary arterial pressure. “+” indicates recommended right heart catheterization. NA = not applicable. * Level of significance, *p* < 0.05.

**Table 3 diagnostics-11-01738-t003:** Characteristics of patients in whom right heart catheterization was performed based on MDT decision.

*n* (%) or Median (IQR)	SSc-PAH + *n* = 7	SSc-PAH − *n* = 9	*p*-Value
Age at diagnosis	62 (51–67)	65 (53–73)	*p* = 0.668
Disease duration	11 (6–20)	5 (4–14)	*p* = 0.235
Sex ratio (M/F)	0	0.5	*p* = 0.192
ACR/EULAR 2013 criteria (*n* [%])	7 (100%)	8 (89%)	*p* = 1.000
Telangiectasia (*n* [%])	7 (100%)	8 (89%)	*p* = 1.000
Right axis on electrocardiogram	2 (29%)	1 (11%)	*p* = 0.550
Anti-centromere antibody (*n* [%])	6 (86%)	2 (22%)	*p* = 0.041 *
NT-pro-BNP (pg/mL)	700 (240–1834)	122 (71–858)	*p* = 0.174
Serum urate (mmol/L)	476 (268–773)	333 (297–470)	*p* = 0.483
WHO-functional classification	3 (2–3)	2 (2–3)	*p* = 0.215
DLCO (% of predicted value)	31 (28–38)	54 (38–61)	*p* = 0.071
FVC (% of predicted value)	82 (69–102)	90 (81–102)	*p* = 0.607
FVC/DLCO ratio	2.63 (2.23–2.93)	2.07 (1.43–2.42)	*p* = 0.012 *
Interstitial lung disease (*n* [%])	2 (29%)	7 (78%)	*p* = 0.126
Tricuspid regurgitation velocity(m/s)	3.2 (3.0–3.6)	3.0 (2.6–3.3)	*p* = 0.208
Estimated sPAP (mmHg)	50 (40–65)	40 (26–49)	*p* = 0.073
Right atrium surface (cm^2^)	19 (16–28)	15 (14–20)	*p* = 0.072
TAPSE (mm)	18 (16–24)	21 (16–22)	*p* = 0.940
Tricuspid S wave (cm/s)	12 (10–15)	14 (10–16)	*p* = 0.691
LVEF (%)	65 (60–65)	65 (60–70)	*p* = 1.000
6MWD (m)	336 (247–501)	444 (419–523)	*p* = 0.295
Pulse oximetry before 6MWT (%)	98 (96–99)	99 (98–100)	*p* = 0.304
Pulse oximetry after 6MWT (%)	88 (79–93)	96 (91–98)	*p* = 0.067
DETECT +	7 (100%)	7 (78%)	NA
ASIG +	7 (100%)	7 (78%)	NA
ESC 2015 +	7 (100%)	6 (67%)	NA
Mean pulmonary arterial pressure (mmHg)	32 (28–39)	18 (18–23)	*p* = 0.002 *
Pulmonary capillary wedge pressure (mmHg)	11 (4–12)	9 (6–13)	*p* = 0.865
Pulmonary vascular resistance (WU)	6.0 (1.9–7.5)	1.5 (1.2–2.2)	*p* = 0.016 *
Cardiac output (L/min)	5.1 (4.0–7.4)	7.1 (5.4–9.1)	*p* = 0.095
Cardiac index (L/min/m^2^)	3.2 (2.1–4.3)	4.1 (3.2–4.3)	*p* = 0.350
PH group I	7 (100%)	0	NA
PH group II	0	2 (22%)	NA
PH group III	0	0	NA
PH group IV	0	0	NA

ACR: American College of Rheumatology; DLCO: diffusing capacity for carbon monoxide; FVC: forced vital capacity; LVEF: left ventricular ejection fraction; NT-pro-BNP: N-terminal pro-B-type natriuretic peptide; PAH: pulmonary arterial hypertension; PH: pulmonary hypertension; sPAP: systolic pulmonary arterial pressure; SSc: systemic sclerosis; TAPSE: Tricuspid Annular Plane Systolic Excursion; WHO: World Health Organization; WU: Wood unit; 6MWD: 6 min walk distance. “+” indicates pulmonary arterial hypertension confirmed by right heart catheterization. * Level of significance, *p* < 0.05; NA = not applicable.

## Data Availability

The data from this study are available upon request from the corresponding author.

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
