# Peer review of "Impact of Three Different Algorithms for the Screening of SSc-PAH and Comparison with the Decisions of a Multidisciplinary Team"

_diagnostics, 2021, doi:10.3390/diagnostics11101738_

Round 1
Reviewer 1 Report
The paper is interesting and well written. I suggest ot the authors to briefly discuss in introduction the role of immune system in the pathogenesis of SSc; particularly the role of VEGF in endothelial dysfunction (see and add as reference paper by Ciprandi et al concerning VEGF in SLE and allergic rhinitis) and the role of Th17 (see and add as reference paper by Murdaca et al concerning Th17 in chronic inflammatory immune-mediated diseases) and, finally, the role of inhibitory NK receptors on CTL (see and add as reference paper by Costa et al published in AIDS 2001).
Author Response
The suggested references have been added in the introduction.
Reviewer 2 Report
The authors propose a study that compares three screening algorithms of pulmonary arterial hypertension in systemic sclerosis (SSc). The topic is interesting and it deserves further scientific interest, however it has a few issues which must be addressed before publication.
Please check English use thorough out the entire manuscript.
Abstract:
The information is not clear, please consult an English specialist and reformulate accordingly.
Introduction:
- Line 47 : Hachulla and colleagues firstly FIRST compared early PAH diagnosis using transthoracic echocardiography
- Authors should stress the importance and novelty of the research.
Materials and Methods
- Was power analysis performed to assess sample size?
- Was randomization applied?
- Callback protocol?
- Line 160: “Quantitative variables were expressed by the median [25th percentile–75th percentile]...”
Line 171: “The median age was 55 [46–66] years old....”
Etc.
All brackets should be round except references so as to not cause confusion.
Discussions:
Line 275: This MDT approach provides an original and practical methods for the screening of SSc-PAH
Line 280: but they may lack of specificity leading to a high number of RHC that can be at risk of adverse
Author Response
The authors propose a study that compares three screening algorithms of pulmonary arterial hypertension in systemic sclerosis (SSc). The topic is interesting and it deserves further scientific interest, however it has a few issues which must be addressed before publication.
Please check English use thorough out the entire manuscript.
Abstract :
The information is not clear, please consult an English specialist and reformulate accordingly.
We have specifically revised english in the abstract :
Abstract: Background. To compare three existing screening algorithms of pulmonary arterial hypertension (PAH) in systemic sclerosis (SSc), with the results of a multidisciplinary team (MDT) meeting from a tertiary center. Methods. We conducted a monocentric longitudinal study from 2015 to 2018. All patients with SSc according to LeRoy’s classification were eligible. Patients were excluded in case of missing data required by any of the 3 screening algorithms. Algorithms were applied for each patient at inclusion. Right heart catheterization (RHC) was performed based on the MDT decision. MDT members were all blinded from the results of the 3 algorithms regarding RHC recommendations. RHC Recommendations of each algorithm were compared with the MDT decision and the impact on diagnosis and management was evaluated. Results. 117 SSc patients were consecutively included in the study, and 99 had follow-up data over the three-year duration of the study (10 deaths). Among the 117 patients, MDT suggested RHC for 16 patients (14%), DETECT algorithm for 28 (24%), ASIG for 48 (41%) and ESC/ERS 2015 for 20 (17%). Among the 16 patients who had RHC, SSc-PAH was diagnosed in 7. Among patients with initial recommendation of RHC based on at least one algorithm, but not according to the MDT meeting, no SSc-PAH was diagnosed during the 3-year follow-up. Results were unchanged when the new 2018 definition of PAH was applied instead of the previous definition. Conclusion. An MDT approach appears interesting for the screening of SSc-PAH, with a significant reduction of RHC performed in comparison with dedicated algorithms. The specific relevance of MDT for the management and follow-up of patients with RHC recommended by existing algorithms but with no PAH warrants further studies.
Introduction :
Line 47 : Hachulla and colleagues firstly FIRST compared early PAH diagnosis using transthoracic echocardiography
This correction has been made.
Authors should stress the importance and novelty of the research.
We have specified in the intro: As SSc-PAH is among the leading causes of SSc-related death, the early diagnosis of SSc-PAH is of high importance. The relevance of MDT in comparison with other screening strategies has never been evaluated to date. To explore this question, we conducted a longitudinal study comparing the baseline number of RHC recommended by the three existing algorithms, and those recommended by an MDT in a tertiary center, and evaluated the baseline prevalence and subsequent incidence of SSc-PAH during a 3-year follow-up to discuss the relevance of each approach.
Materials and Methods
Was power analysis performed to assess sample size ?
No power analysis was performed to assess the sample size. All patients who met inclusion criteria during the study period were included.
Was randomization applied ?
No randomization was applied. All algorithms have been applied for each patient.
Callback protocol ?
There was no callback protocole.
Line 160 : “Quantitative variables were expressed by the median [25th percentile–75th percentile]…”
Line 171 : “The median age was 55 [46–66] years old….”
Etc.
All brackets should be round except references so as to not cause confusion.
Brackets are modified as suggested.
Discussions :
Line 275 : This MDT approach provides an original and practical methods for the screening of SSc-PAH
This correction is made (method instead of methods).
Line 280 : but they may lack of specificity leading to a high number of RHC that can be at risk of adverse
This correction is made (lack specificity instead of lack of specificity).